# INDETERMINATE DOMAIN Transcription Factors in Crops: Plant Architecture, Disease Resistance, Stress Response, Flowering, and More

**DOI:** 10.3390/ijms251910277

**Published:** 2024-09-24

**Authors:** Akiko Kozaki

**Affiliations:** 1Graduate School of Science and Technology, Shizuoka University, Ohya 836, Suruga-ku, Shizuoka 422-8021, Japan; kozaki.akiko@shizuoka.ac.jp; 2Department of Biological Science, Faculty of Science, Shizuoka University, Ohya 836, Suruga-ku, Shizuoka 422-8021, Japan; 3Course of Bioscience, Department of Science, Graduate School of Integrated Science and Technology, Shizuoka University, Ohya 836, Suruga-ku, Shizuoka 422-8021, Japan

**Keywords:** *INDETERMINATE DOMAIN* (*IDD*) transcription factor, rice and maize, agriculture, C_3_ and C4 plant, leaf differentiation, plant architecture, disease resistance, stress response, flowering, seed development

## Abstract

*INDETERMINATE DOMAIN* (*IDD*) genes encode plant-specific transcription factors containing a conserved IDD domain with four zinc finger motifs. Previous studies on *Arabidopsis IDDs* (*AtIDDs*) have demonstrated that these genes play roles in diverse physiological and developmental processes, including plant architecture, seed and root development, flowering, stress responses, and hormone signaling. Recent studies have revealed important functions of *IDDs* from rice and maize, especially in regulating leaf differentiation, which is related to the evolution of C_4_ leaves from C_3_ leaves. Moreover, *IDDs* in crops are involved in the regulation of agriculturally important traits, including disease and stress resistance, seed development, and flowering. Thus, *IDDs* are valuable targets for breeding manipulation. This review explores the role of *IDDs* in plant development, environmental responses, and evolution, which provides idea for agricultural application.

## 1. Introduction

Due to recent climate change, reduced availability of water and energy resources, and a growing population, it is urgent to develop high-yield crops that can thrive in areas affected by high temperatures and drought. *INDETERMINATE DOMAIN* (*IDD*) genes, which encode plant-specific transcription factors, are emerging as promising target candidates for breeding.

Maize *INDETERMINATE 1* (*ZmID1*), which regulates the transition from vegetative to reproductive growth, was the first characterized *IDD* gene [1]. Comparative genomic analyses have revealed that it belongs to a highly conserved transcription factor family [2,3]. The *IDD* gene family encodes transcription factors containing a conserved IDD domain with two canonical C_2_H_2_ zinc fingers (ZFs) and two C_2_HC ZFs in the N-terminal region [1,2]. ZmID1 is localized to the nucleus and binds an 11 bp DNA consensus sequence 5′-TTTTGTC(G/C)(T/C)T/a)T/a)T-3′, which many IDD proteins can bind [3,4,5,6,7,8]. Structural and biochemical analyses have shown that IDDs bind to DNA via ZF1-ZF3 [9].

Bioinformatics analysis has indicated that *IDDs* arose from a common ancestor of *Streptophyta* [10]. *IDDs* duplicated extensively after plants colonized land, resulting in ten lineages. The functions of *IDDs* have diversified with the evolution of terrestrial plants. Since the emergence of *IDD* genes in land plants, they have been divided primarily into two lineages: SG5 (including *Arabidopsis AtIDD14*, *15*, and *16*) and others. The SG5 lineage exists in all land plants. Although the remaining *IDD* lineages diversified during the evolution of land plants, the MGP lineage was retained in all land plants [7,10]. Moreover, although the number of *IDD* genes is similar in monocot and dicot genomes, monocots have a greater number of sequence clusters, most of which are grass-specific [10].

The functions of the *IDD* genes have mainly been clarified in *Arabidopsis*. AtIDD14, 15, and 16 cooperatively regulate auxin biosynthesis and transport, resulting in the modulation of aerial organ morphogenesis and gravitropic responses [11,12].

AtIDD2 (GAF1), AtIDD3 (MGP), and other AtIDDs interact with DELLA proteins, which are negative regulators of Gibberellic acid (GA) signaling, to regulate genes involved in GA synthesis (*GA3ox1* and *GA20ox1*) and GA signaling (*SCL3*) [5,6,13].

AtIDD10 (JKD) and MGP interact with the complex of SCARECRAW (SCR)-SHORT ROOT (SHR), GRAS family transcription factors involved in the radial pattern formation of root ground tissues and regulate target genes including *SCR* [14,15,16]. Other AtIDDs (AtIDD4, 6, 8 and 9) are also involved in the control of tissue formation in root development [17,18].

Although functional information on IDDs other than *Arabidopsis* is limited, recent studies have revealed several important functions of IDDs in crops. This review focuses on the biological functions and mechanisms of *IDDs* in crops. Because the SG5 lineage is somewhat distant from the remaining groups of IDDs [10], these groups are discussed separately.

## 2. Functions of *IDD* SG5 Lineage Genes

### 2.1. Leaf Differentiation

Leaves of C_4_ plants usually more efficiently use higher radiation, water, and nitrogen for photosynthesis than those of C_3_ plants. Engineering C_4_ traits into C_3_ crops can substantially increase crop yield, especially under hot and arid conditions [19,20,21]. To achieve this, it is important to understand the molecular mechanisms underlying the development of anatomical traits in C_4_ plants.

Two rice *IDDs*, *OsIDD12* and *OsIDD13*, are involved in regulating venous differentiation [22]. In the early stages of leaf development, cell division in the ground meristem produces either mesophyll cells or vascular initials [23,24]. Parallel venation of leaf veins is a characteristic feature of grasses, and the pattern of venation differs between C_3_ and C_4_ plants. The leaves of C_4_ plants possess high vein density and two photosynthetic cell types associated with C_4_ photosynthesis. In both C_3_ and C_4_ leaves, there are midveins, large lateral veins, and intermediate longitudinal veins. However, C_4_ leaves produce an additional file of small longitudinal veins during leaf expansion, resulting in decreased distance between the veins [25,26,27] (Figure 1).

In rice plants, OsSHR1 and OsSHR2 have been shown to act redundantly to promote mesophyll cell identity and to determine the positioning of minor veins [22]. Moreover, OsSHR1 and OsSHR2 promote the development of sclerenchyma cells and repress the differentiation of bulliform cells on the abaxial surfaces of the minor veins [22].

Liu et al. (2023) [22] have shown that OsSHRs interact with OsIDD12 and OsIDD13 to regulate vein patterning in rice plants. Double mutants of *OsSHR1* and *OsSHR2* (*Osshr1*, *Osshr2*) and *OsIDD12* and *OsIDD13* (*Osidd12 Osidd13*) show similar phenotypes: reduced mesophyll cell numbers, increased minor vein numbers, reduced lobing of mesophyll cells, the appearance of morphologically distinct minor veins, and thickened cell walls of vascular bundles [22]. These traits support the C_4_ anatomy.

OsIDD12 and OsIDD13 bind directly to the IDD-binding sequence located in intron 3 of the rice *PIN FORMED5c* (*OsPIN5c*) and suppress *OsPIN5c* expression in combination with OsSHRs (Figure 2a). These results indicate that the SHR-IDD-PIN5c circuit is an ideal target for optimizing anatomical traits in rice to support C_4_ biochemistry [22].

In barley, *BROAD LEAF1* (*HvBLF1*) encodes an IDD protein corresponding to OsIDD12 and OsIDD13. Mutations in *HvBLF1* form wider but slightly shorter leaves owing to changes in the number of longitudinal cell files [28]. Mutations in maize *IDD14* (*ZmIDD14*) and *ZmIDD15*, which are orthologs of *OsIDD12* and *OsIDD13*, also show similar leaf phenotype [29]. Therefore, this phenotype is a common feature among *Osidd12 Osidd13*, *Zmidd14 Zmidd15*, and *Hvblf1* mutants.

### 2.2. Plant Architecture

Tiller angle, defined as the angle between the main culm and its side tillers, is an important trait for breeding because it affects plant architecture [30]. In cultivated rice, neither a compact nor spread-out plant architecture is beneficial for grain production [31]. Whereas plants with a spread-out architecture can escape diseases that are increased by high humidity, the photosynthetic efficiency and grain yield per unit area decrease because they occupy more space, and shading and lodging are increased. However, whereas compact plants are suitable for high-planting densities, light capture is not efficient, and they are more susceptible to pathogens and insects transmitted by contact. Thus, an appropriate tiller angle is essential for rice production [31]. Besides the tiller angle, lamina inclination (leaf angle) is also an important trait because it affects the amount of light that the leaf can capture for photosynthesis. In lamina inclination, auxin and brassinosteroids (BRs) are involved [32,33,34,35].

The rice mutant of *LOOSE PLANT ARCHITECTURE1* (*LPA1*, *OsIDD14*) displays a relatively large tiller and leaf angle. *LPA1* also affects shoot gravitropism [36]. *LPA1* is a functional ortholog of *Arabidopsis IDD15* (*AtIDD15*/*SHOOT GRAVITROPIAM5* (*SGR5*)), which shows reduced gravitropism [12]. In *Arabidopsis*, *AtIDD14*, *AtIDD15*, and *AtIDD16* cooperatively regulate the lateral organ morphogenesis and gravitropism via auxin biosynthesis and transport. Further analysis indicated that these IDD proteins promote auxin biosynthesis and transport by regulating spatial auxin accumulation by directly targeting *YUCCA5* (*YUC5*), *TRYPTOPHAN AMINOTRANSFERASE OF ARABIDOPSOS* (*TAA1*), and *PIN1* [11].

Liu et al. (2016) showed that LPA1 suppresses auxin signaling by interacting with C-22-hydroxylated or 6-deoxo BRs, which regulate lamina inclination independent of OsBRI1, a BR receptor [37]. They also showed that LPA1 influences the expression of three *OsPIN* genes (*OsPIN1a*, *PIN1c*, and *PIN3a*) [37] (Figure 2c).

The mutant of maize *LPA1* (*ZmLPA1*, *ZmIDD16*) shows a phenotype similar to *Oslpa1* [38,39].

Mutations in *ZmIDD14* and *ZmIDD15* decreased the leaf angle by reducing adaxial sclerenchyma thickness and increasing the number of colorless cell layers [29]. Moreover, the double mutant of *ZmIDD14* and *ZmIDD15* (*Zmidd14 Zmidd15*) exhibited an asymmetrically smaller auricle, which might be the result of a failure to maintain the expression of *LIGULELESS* (*LG1*), the key gene controlling auricle size. ZmIDD14 and ZmIDD15 interact with the INCREASED LEAF INCLINATION1 (ZmILI1), a bHLH-type protein, which binds to the promoter of *LG1* to regulate the gene [40] (Figure 2d). The plant architecture of the double mutant *Zmidd14 Zmidd15* has an advantage under high-density conditions, and grain yield is increased under high planting density [29].

### 2.3. Stress Tolerance

To thrive, sessile plants have developed a range of strategies to manage challenging environmental conditions, including drought, salinity, and extreme temperatures.

Although the spread-out phenotype of *lpa1* does not seem ideal for breeding, *lpa1* shows advantages under drought conditions in dwarf rice [41]. *LPA1* is expressed in the pre-vascular cells of leaf primordia and plays a role in the metaxylem enlargement of aerial organs by regulating the genes involved in carbohydrate metabolism and cell enlargement. The narrow metaxylem of *lpa1* plants exhibits efficient water use and drought tolerance. Under the genetic background of a semi-dwarf (*dense and erect panicle1-ko* (*dep1-ko*) or *ebisu dwarf* (*d2*)), *lpa1* shows optimal water supply and drought resistance without affecting the grain-filling rate [41].

### 2.4. Disease Resistance

Since plant diseases reduce crop yield, producing crops that are resistant to pathogens is an important objective in plant breeding. However, the antagonistic relationship between crop yield and immune pathways makes breeding difficult [42].

Sheath blight disease (ShB) is one of the most devastating diseases caused by *Rhizoctonia solani* in rice [43]. In rice plants, overexpression of *OsLPA1* results in resistance to ShB via the activation of *PIN1a* expression (Figure 2e) [44]. OsLPA1 binds to the *OsPIN1* promoter and activates gene expression. Furthermore, 3-indole acetic acid (IAA) treatment and *PIN1a* overexpression enhances rice resistance to ShB, indicating that auxin enhances resistance to ShB. The levels of pathogen resistance in genes *PBZ1* and *PR1b* were higher in *LPA1* or *PIN1* overexpression lines and lower in *PIN1a* RNAi lines than in the WT when plants were infected with *Rhizocotonia sloani AG1-1A.* These results indicate that *LPA1* might control auxin transport via the regulation of *PIN1a* expression to increase planting density and activate plant defense gene expressions [44].

Subsequent research identified several factors that interact with OsLPA1, OsIDD13, OsIDD3, kinesin-like protein (KLP), and DENSE AND PANICLE 1 (DEP1) [45,46,47]. OsIDD13 and OsIDD3 positively and negatively regulate *OsPIN1a* expression, respectively [45]. Moreover, OsIDD13, OsIDD3 and OsLPA1 form a transcription factor complex that regulates the *OsPIN1a* gene. Accordingly, OsIDD13 and OsIDD3 increase and decrease ShB resistance, respectively [45] (Figure 2e).

KLP promotes rice resistance to ShB by enhancing the expression of *OsPIN1a* together with OsLPA1 [46]. DEP1 negatively regulates rice resistance to ShB by interacting with OsLPA1 and inhibiting the DNA-binding ability of OsLPA1 to reduce *OsPIN1a* [47] (Figure 2e).

### 2.5. Fruit Shapes

A genome-wide association study (GWAS) of tomato plants identified the *POINTED TIP* (*PT*) gene, which regulates the protuberance of tomato fruit tip [48]. *PT* encodes the IDD protein that is classified as the SG5 group. A single-nucleotide polymorphism alters histidine (H) to arginine (R) in one of the zinc fingers of the ID domain (referred to as PT^R^). In this context, PT^H^, with an intact zinc finger, promotes fruit development without a pointed tip by downregulating *FRUITFULL2* (*FUL2*), which modifies auxin transport. Conversely, RT^R^ is unable to suppress *FUL2* expression, leading to the formation of a pointed fruit tip [48] (Figure 2g).

The regulation of auxin synthesis and transport is a shared function among SG5 lineage IDDs in *Arabidopsis* and grasses [11,22,38]. However, several functions of the SG5 lineage IDDs from *Arabidopsis* have not been reported in rice or maize.

AtIDD14 regulates starch metabolism by directly activating the promoter of the *Qua-Quine Starch* (*QQS*) gene. An alternatively spliced variant of AtIDD14 is induced by cold temperatures, and competitively inhibits AtIDD14 activity by forming a heterodimer, and as a result, starch accumulation is modified in response to cold [49].

AtIDD14 regulates ABA-mediated drought tolerance by promoting ABA sensitivity and ABA-mediated stomatal closure by interacting with the bZIP-type transcription factors ABFs/AREBs [50].

AtIDD16 negatively regulates stomatal initiation by directly binding to and suppressing the *SPEECHLESS* promoter [51]. Although the combination of OsSCR and OsSHR has been reported to control stomatal development in rice [22,52], the involvement of OsIDDs in this process remains unclear.

## 3. Function of *IDD* Genes in Lineages Other than SG5

### 3.1. Stem Elongation, Secondary Cell Wall

Crop height is an important breeding trait. GAs are crucial for various aspects of plant growth and development, including stem elongation, seed germination, development, and flowering [53]. In the absence of GAs, DELLA proteins repress various GA responses in plants. However, in the presence of GAs, DELLA proteins are degraded through the 26S-proteasome pathway, which eliminates their inhibitory effects, and various GA-dependent responses occur [54].

Rice *GROWTH-REGULATING FACTOR1* (*OsGRF1*) is a GA-responsive gene expressed mainly in the intercalary meristems of rice internodes [55]. The GRF family has been identified as a conserved family of plant-specific transcription factors in various plant species [56,57,58]. GRFs have diverse functions in plant development and growth, including regulation of GA biosynthesis and stem elongation [57,59]. In *Arabidopsis* and rice, *GRFs* are the targets of miR396 [60,61,62], a highly conserved microRNA family found in all land plants [63].

OsIDD2 binds directly to the *OsmiR396a* promoter and enhances its expression by interacting with SLR1 (rice DELLA) [64]. In the absence of GA, OsIDD2 and higher levels of SLR1 promote *OsmiR396* expression, resulting in the suppression of *OsGRFs*. In contrast, the mRNA levels of *OsGRFs* increase in the presence of GA because of the degradation of SLR1 and reduction in the level of *miR396*, resulting in stem elongation [64] (Figure 2h).

In addition to dwarfism, *OsIDD2* overexpressing plants exhibit fragile leaves and reduced lignin content, which are characteristics commonly observed in plants with defects in secondary cell wall formation [8]. *OsIDD2* negatively regulates the transcription of genes involved in lignin biosynthesis, including cinnamyl alcohol dehydrogenase 2 and 3 (*CAD2* and *CAD3*), as well as sucrose metabolism, including *sucrose synthase 5* (*SUS5*). Of these, *CAD2* and *CAD3* are directly regulated by OsIDD2 [8] (Figure 2h). Collectively, these findings suggest that OsIDD2 plays a negative role in secondary cell wall formation [8] and stem elongation [64].

### 3.2. Nitrogen Metabolism

The mutant of *OsIDD10* exhibits ammonium hypersensitivity during root growth with root tip coiling [65]. OsIDD10 directly activates *Ammonium transporter 1:2* (*AMT1;2*) and *Glutamate dehydrogenase2* (*GDH2*) expression by binding to their promoters and modulating NH_4_^+^ uptake in rice, depending on N supply [65] (Figure 2i). In the ammonium-dependent regulation of root growth by OsIDD10, *CALCINEURIN B-LIKE INTERACTING PROTEIN KINASE 9* (*CIPK9*) has been identified as a direct target of OsIDD10 [66] (Figure 2i).

### 3.3. Seed Development

The cereal endosperm stores nutrients that provide energy for germination and early seedling development and is important for human food, livestock feed, and industrial commodities.

In Maize, the duplicated genes *naked endosperm1* (*nkd1*) and *nkd2* encode ZmIDDveg9 and ZmIDD9, respectively. Double mutants of *NKD1* and *NKD2* show various effects, including multiple layers of peripheral endosperm, opaque and floury endosperm texture, decreased anthocyanin and carotenoid accumulation, decreased kernel dry weight, and occasional vivipary [67,68].

Normal maize has a single aleurone layer. Meanwhile, the *naked* double mutant produces multiple outer cell layers of partially differentiated cells that occasionally express aleurone identity markers, such as *Viviprous1* (*VP1*) [67,68]. NKD1 and NKD2 directly regulate gene expression, including *Opaque2* (*O2*), *VP1*, and *Prolamin-box-binding factor1* (*PBF1*), while NKD2 negatively regulates *NKD1* expression, indicating feedback regulation [68] (Figure 2j).

Subsequent studies have shown that *NKD1*, *NKD2* and *O2* interact to affect endosperm development [69]. These three factors promote starch metabolism, lipid storage, and storage protein accumulation and constrain the hormone response, cell wall organization, and other cellular developmental processes during the transition from cellular development to storage compound accumulation [69].

In *Arabidopsis*, ectopic expression of *AtIDD1*/*ENHYDROUS* (*ENY*) disrupts seed development and delays endosperm depletion and testa senescence, resulting in an abbreviated maturation program [70].

### 3.4. Leaf Differentiation

Recent research has revealed that *NKDs* in functional combination with *SCR* control the number of mesophyll cells specified between veins in the leaves of C_4_ grass [71].

The mutation of *SCR* genes in maize leads to disruption in inner leaf patterning, including veins that are separated by only one mesophyll cell, an increase in sclerenchyma, and the presence of ectopic bundle sheath cells; however, no phenotypic changes in stomata are observed. In contrast, mutations in *SCR* in rice show no stomata, but no phenotype in inner leaf patterning [72]. Furthermore, mutations in the *SCR* gene in *Setaria viridis*, another C_4_ plant, result in a mixed phenotype of maize and rice; veins are separated by only one mesophyll cell and no stomata. Loss of *NKD* shows no obvious perturbation in leaf development in maize, *Setaria*, or rice (rice *NKD* corresponds to *OsIDD10*). However, *scr:nkd* mutants in maize and *Setaria*, but not in rice, exhibit an increased proportion of fused veins with no intervening mesophyll cells [71] (Figure 2b).

These results indicate that the ancestral role of *SCR* in grass leaves was the patterning of epidermal cell types, and as C_4_ grasses evolved, *SCR* collaborated with *NKD* to regulate the pattern of inner leaf cell types. Some C_4_ species such as *Setaria* have retained their ancestral function, whereas others such as maize have lost it [71].

### 3.5. Disease and Stress Resistance

As mentioned above, *OsIDD3* negatively regulates the defense against ShB by suppressing auxin signaling and activating BR signaling [45,73]. *OsIDD3* expression is induced by inoculation with *Rizoctonia solani* and exogenous auxins. *OsIDD3* overexpressing plants developed a wider tiller angle and exhibit altered shoot gravitropism, whereas the knockout mutants show no visible phenotype. OsIDD3 binds directly to the *PIN1b* promoter and represses its expression [73].

Moreover, *OsIDD3* is directly regulated by ABI3/VP1-like 1 (RAVL1) and positively regulates the expression of *BRI1*, a BR receptor gene, and *D2* and *D11*, the BR biosynthesis genes, in an indirect manner, resulting in the activation of BR signaling [74] (Figure 2e).

On the other hand, OsIDD3 enhances chilling tolerance by activating the *C-repeat binding factor 1* (*CBF1*) expression by binding to the promoter of *CBF1* even though the expression of *OsIDD3* is not affected by cold or ABA stimuli [75] (Figure 2k).

*OsIDD10* is also involved in pathogen resistance. OsIDD10 interacts with BRASSINAZOLE-RESISTANT1 (BZR1), a key transcription factor in BR signaling and activates *AMT1;2* directly [76]. Under light conditions, Phytochrome B (PhyB) interacts with OsIDD10 and BZR1 to inhibit their DNA-binding activities, resulting in reduced *AMT1;2* expression. Under dark conditions, OsIDD10 and BZR1 are released from PhyB and enhance NH_4_^+^ uptake by activating *AMT1;2*. The phyB mutant exhibits tolerance to ShB and saline–alkaline stress because of the expression of *AMT1;2*, which enhances NH_4_^+^ uptake, and tolerance to ShB and saline–alkaline stresses is increased. Further experiments have demonstrated that PhyB-OsIDD10-AMT1;2 signaling regulates the saline–alkaline response, whereas the PhyB-BZR1-AMT1;2 pathway modulates ShB resistance [76] (Figure 2f,l).

In contrast, OsIDD10 negatively regulates rice resistance to ShB by interacting with NAC079 to inhibit ethylene biosynthesis and signaling genes [77]. Exogenous ethylene induces the expression of pathogenesis-related genes (*PR* genes) in rice [78]. Overexpression of the ethylene biosynthesis gene, *OsACS2*, which encodes 1-amino cyclopropane-1-carboxylic acid synthase, enhances ShB resistance [79]. The OsIDD10 and NAC079 complex directly activates *ETR2*, a negative regulator of ethylene signaling. CALCINEURIN B-LIKE INTERACTING PROTEIN KINASE 31 (CIPK31) interacts with phosphorylate NAC079 to increase its transcriptional activity. In addition, AMT1 inhibits the expression of *OsIDD10* and *CIPK31*, resulting in activation of the ethylene signaling pathway, which positively regulates ShB resistance [77] (Figure 2f).

Genome-wide characterization of *OsIDDs* shows that the expressions of most *OsIDD* genes respond to abiotic stresses, such as low temperature and drought, and plant hormones, such as auxin, GA, and ABA, indicating that many *OsIDDs* are involved in stress responses [80].

In *Arabidopsis*, *AtIDD4* negatively regulates the basal immune response and pathogen-associated molecular pattern (PAMP)-triggered immunity [81]. AtIDD4 directly binds to and activates the promoter of the *SALICYLIC ACID GLUCOSYLTRANSFERASE1* (*SAGT1*) encoding enzyme, which converts salicylic acid (SA) to the biologically inactive storage forms SAG and SGE [81,82]. According to the classification of *IDDs* by Prochetto and Reinheimer (2020), *ZmNKDs*, *OsIDD3*, and *AtIDD4* belong to the same clade [10].

### 3.6. Flowering

In the plant life cycle, the transition from vegetative to reproductive growth is a crucial event, and it is fine-tuned by both environmental and endogenous factors [83,84]. The timing of rice flowering, or heading date, is a critical agricultural characteristic that affects rice yield and adaptation to various climatic and environmental conditions [85,86].

The first identified *IDD* was *ZmID1*, which regulates flowering [1]. *ZmID1* is expressed in immature leaves, and its expression is not affected by variations in light, dark, or circadian patterns [87]. *ZmID1* regulates the expression of *CENTRORADOALIS 8*, (*ZCN8*) encoding FT homologous protein which possesses florigenic activity [88] (Figure 2m). However, whether ZCN8 is directly regulated by ZmID1 has not yet been determined.

In rice, *RICE INDETERMINATE1* (*RID1*)/*Early heading date 2* (*Ehd2*)/*OsID1*/*Ghd10* has been identified as an ortholog of *ZmID1* (hereafter referred to as *RID1*) [89,90,91]. Mutations of *RID1* exhibit a never-flowering phenotype. *OsIDD4* has been identified as a *Suppressor of rid1* (*SID1*) that rescues the never-flowering phenotype of *the rid1* mutant [92]. The *sid1* mutant displays a moderately late flowering phenotype. In addition to *SID1*, *OsIDD1* and *OsIDD6* have been found to restore *rid1* to flowering when overexpressed. These results indicate that these *IDDs* redundantly regulate rice flowering. Moreover, RID1 and SID1 directly target the promoter regions of *Heading date 3a* (*Hd3a*: rice *FT*) and *FLOWERING LOCUS T1* (*RFT1*) [92].

Recent studies have shown that RID1 interacts with the methyltransferases SET DOMAIN GROUP PROTEIN 722 (SDF722) and SLR1 [93]. Zhang et al. (2022) proposed the following model: When the *SLR1* level is high, *Hd3a* and *RFT1* expressions are suppressed because RID1 is inactivated by interacting with SLR1. During rice development, SLR1 levels progressively decrease in young leaves, leading to the reduced suppression of RID1 and SDG722. The RID1 released from SLR1 recruits SDG722 to promote the accumulation of H3K4me3 and H3K36me3 in the chromatin regions of *Hd3a* and *RFT1*. As a result, the chromatin states of *Hd3a* and *RFT1* in the newly developed leaves become activated, allowing them to be recognized and activated by upstream flowering genes. The accumulation of sufficient amounts of florigen proteins (Hd3a and RFT) in the shoot apical meristem initiates a shift from the vegetative to the reproductive stage [93] (Figure 2m).

Analysis of the genome-wide binding sites of RID1 reveal that RID1 binds to the TTTGTC motif and significantly enriches the TEOSINTE BRANCHED1, CYCLOIDEA, PCF (TCP), bHLH, and SQUAMOSA PROMOTER BINDING PROTEIN (SBP) binding motifs by interacting with the novel flowering regulators OsTCP11, OsPIL12, and OsSPL14, respectively [93]. ChIP-seq results indicate that RID1 binds to the promoters of several flowering genes, including *HD1* (rice *CO*), *GRAIN NUMBER*, *PLANT HEIGHT AND HEADING DATE* (*GHD*), and so on. Moreover, a novel RID1 target, *OsERF#136*, which encodes the AP2 transcription factor, has been identified as a flowering repressor. RID1 negatively regulates the expression of OsERF#136, resulting in flower induction [94] (Figure 2m).

Recently, ZmID1 orthologs have been identified in *Brachypodium distachyon* (*BdID1*) and *Sorghun bicolor* (*SbID1*), and mutations in these genes result in delayed flowering, as in maize and rice, by regulating several flowering genes, including the homologs of *FT* and *CO* [95,96]. The orthologs of *ZmID1* belong to a grass-specific lineage of *IDDs* [2,10]

In *Arabidopsis*, *AtIDD8* promotes photoperiodic flowering by directly activating the expression of *SUCROSE SYNTHASE4* (*SUS4*), thus regulating sugar transport and metabolism [97]. Under energy-deprived conditions, the SUCROSE NONFERMENTING-1-RELATED PROTEIN KINASE1 (SnRK1) is activated [98] and KIN10, the subunits of SnRK1, interact with and phosphorylate two serin residues of AtIDD8 to repress its activity [99].

## 4. Concluding Remarks and Future Perspectives

Recent studies have revealed several important functions of *IDD* genes in rice and maize. The functions of IDD genes discussed in this review are summarized in Table 1. It is particularly interesting that OsIDD12 and OsIDD13 regulate vein density by interacting with OsSHR, indicating that these transcription factors contributed to the evolution of C_4_ plants from C_3_ plants [22]. However, ZmIDD14 and 15, which correspond to OsIDD12 and 13, are involved in the regulation of the leaf angle [28].

Instead, the functional combination of *NKD* and *ZmSCR* was involved in the regulation of the venation patterns in the inner leaf tissues of C_4_ plants, but not in those of C_3_ plants [71]. Although both rice and maize belong to *Poaceae* family, some functions of *IDDs* vary between C_3_ rice and C_4_ maize plants, indicating *IDDs* have changed their functions during evolution from C_3_ to C_4_ plants.

Although some functions are conserved between Arabidopsis (dicot) and rice (monocot), many functions have been reported only in either *Arabidopsis* or rice. In *Arabidopsis*, *IDDs* have been extensively analyzed in root development, GA synthesis, and signaling [27,100]. However, there are few reports on these functions in rice or maize *IDDs*. Further analyses will reveal the undiscovered functions of *IDDs* in rice and maize.

The function of *IDDs* in crops other than rice and maize remain largely unknown. Recent genome-wide studies have identified *IDD* genes in several plants and analyzed their structure and expression [101,102,103]. Further detailed functional analyses of these genes and *IDD* genes in several other crops are required.

As outlined in this review, *IDD* genes regulate agriculturally important traits, indicating that they are valuable targets for genetic modifications to optimize plant growth and development under various environmental conditions.

## Figures and Tables

**Figure 1 ijms-25-10277-f001:**
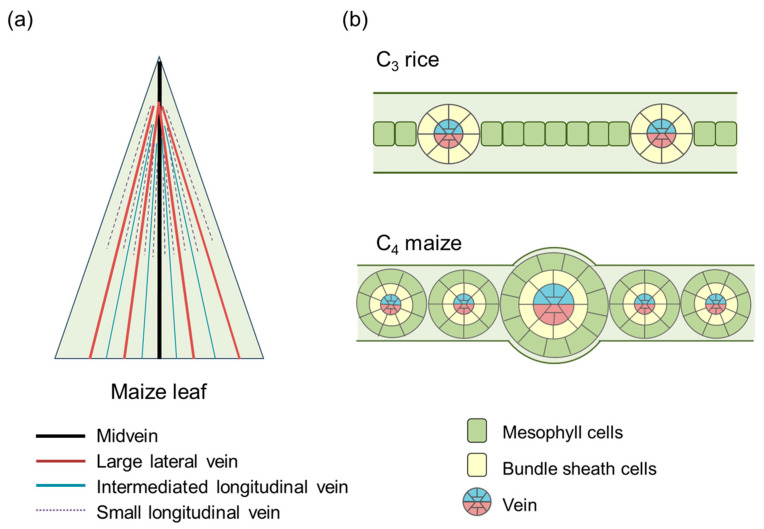
Leaf structure of C_4_ maize and C_3_ rice (based on [27]). (**a**) Schematic representation of C_4_ maize leaf. After midvein and large lateral veins are established, minor veins and transverse veins are formed. Transvers veins, which connect longitudinal parallel veins, are not included in this figure. (**b**) Schematic representation of cross-sections of leaves from C_3_ rice and C_4_ maize. Between two adjacent veins, there are two mesophyll cells in C_4_ maize, while many more mesophyll cells are present in C_3_ rice.

**Figure 2 ijms-25-10277-f002:**
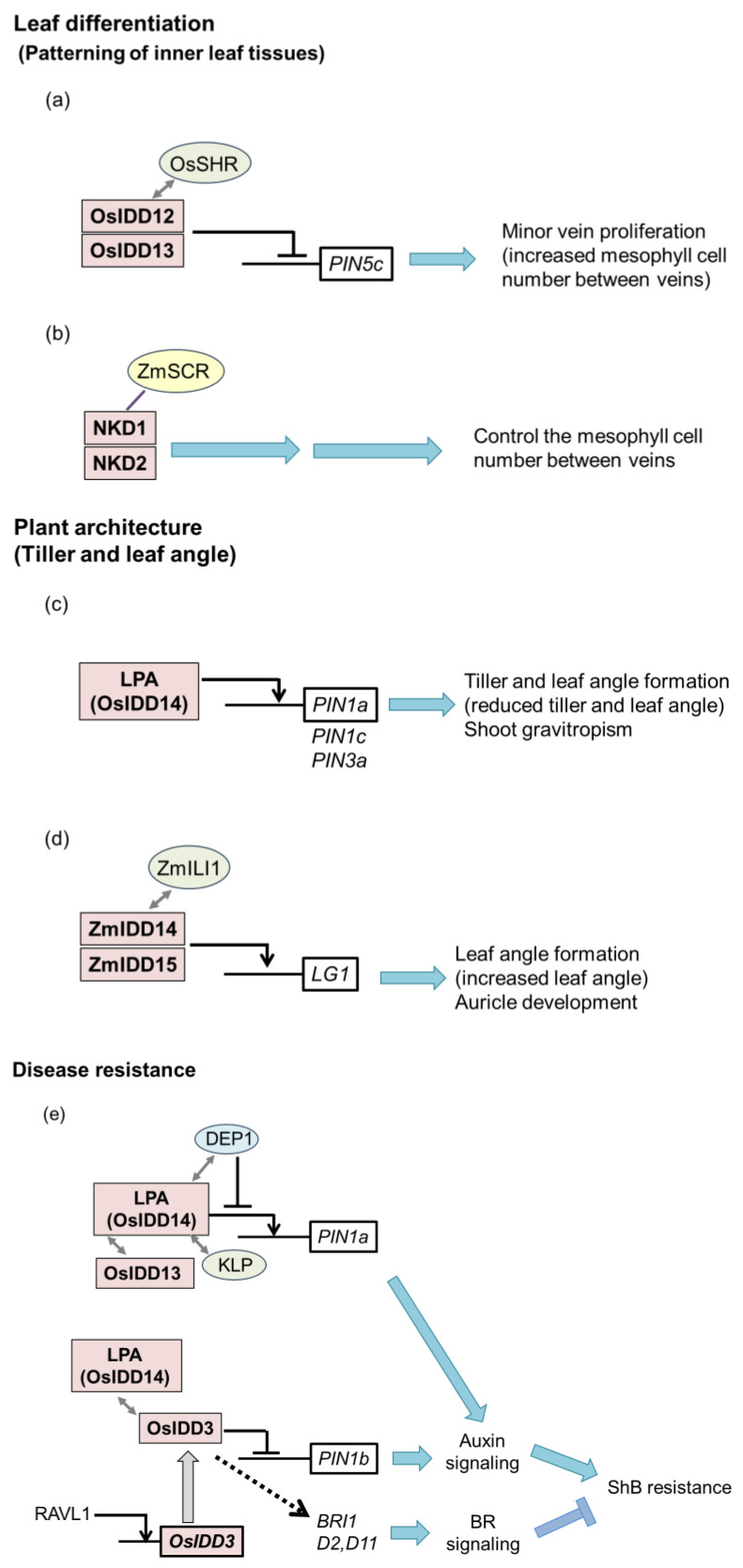
Functions of IDD transcription factors in various aspects of physiological and developmental processes. Schematic diagrams of IDD functions in leaf differentiation (**a**,**b**), plant architecture (**c**,**d**), disease resistance (**e**,**f**), fruit shape (**g**), stem elongation, secondary cell wall (**h**), nitrogen metabolism (**i**), seed development (**j**), stress response (**k**,**l**), and flowering (**m**). The arrows and T-shaped lines represent positive and negative regulation, respectively. The direct target genes of IDDs are in open boxes, and the left-side lines of the boxes indicate promoters. Dashed arrow indicates indirect activation. Double-headed arrows indicate protein–protein interaction. The red cross in (**g**) indicates that PT^R^ cannot function as a suppressor.

**Table 1 ijms-25-10277-t001:** *IDD* gene functions described in this study.

Gene	ID	Function	Lineages (Based on [10])	References
Rice				
*OsIDD1*	LOC_Os03g10140	Flowering transition	GAF1	[92]
*OsIDD2*	LOC_Os01g09850	Stem elongation		[64]
		Secondary cell wall structure		[8]
*OsIDD3*	LOC_Os09g38340	Disease resistance	NKD	[45,73]
		Chilling torelance		[75]
		BR signaling		[74]
		Auxin transport		[73]
*OsIDD4* (*SID1*)	LOC_Os02g45054	Flowering transition	NKD	[92]
*OsIDD6*	LOC_Os08g44050	Flowering transition	NKD	[92]
*OsIDD10* (*OsNKD*)	LOC_Os04g47860	N metabolism	NKD	[65,66]
		Disease resistance		[76,77]
		Saline alkaline tolerance		[76]
*OsIDD12*	LOC_Os08g36390	Leaf vein formation	SG5	[22]
*OsIDD13*	LOC_Os09g27650	Leaf vein formation	SG5	[22]
		Disease resistance		[45]
*OsIDD14* (*LPA*)	LOC_Os03g13400	Leaf and tiller angle formation	SG5	[36,37]
		Disease resistance		[44,45,46,47]
		Auxin transport		[36,37,44,45,46,47]
*OsID1* (*RID1*)	LOC_Os10g28330	Flowering transition	ID1	[89,90,91]
Maize				
*ZmIDDveg9* (*NKD1*)/*Zmveg9* (*NKD2*)		Seed (endosperm) deveopment	NKD	[67,68]
		Leaf vein formation		[71]
*ZmIDD14*/*ZmIDD15*		Leaf angle formation/Auricle development	SG5	[29]
		Leaf formation		[29]
*ZmIDD16* (*ZmLPA1*)		Leaf and tiller angle formation	SG5	[38,39]
*ZmID1*		Flowering transition	ID1	[1,87]
Barley				
*BROAD LEAF1* (*HvBLF1*)		Leaf and tiller angle formation	SG5	[28]
Tomato				
*PT*		Fruit shapes	SG5	[48]

## Data Availability

Not applicable.

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
