# Peer review of "INDETERMINATE DOMAIN Transcription Factors in Crops: Plant Architecture, Disease Resistance, Stress Response, Flowering, and More"

_ijms, 2024, doi:10.3390/ijms251910277_

Round 1

Reviewer 1 Report

Comments and Suggestions for Authors

The overall organization and logic of the article are good, but there are a few points that may need modification and enhancement:

1. The last sentence of the abstract mentions that the study offers fundamental insights for agricultural applications. (Line 22) However, this aspect is barely covered in the article.

2. The article discusses the functions of IDD genes in different species. It is recommended to summarize these genes and their functions in a table, which would provide readers with a clearer understanding and also align with Figure 2.

3. Can you provide a phylogenetic tree of IDD protein from different species mentioned in the article? This would help readers better visualize the evolutionary relationships between these genes.

4. In the section Function of IDD genes in SG5 lineage, “leaf differentiation” is discussed first. However, in the section Function of IDD genes in other lineages than SG5, “leaf differentiation” is not addressed first. It is recommended to maintain consistency between the two sections. In the Function of IDD genes in SG5 lineage section, Plant architecture mainly refers to the tiller and leaf angle. In contrast, in the Function of IDD genes in other lineages than SG5, stem elongation is also considered part of plant architecture, but it is referred to specifically as stem development. For better logical consistency, it is advisable to use the same terminology and structure in both sections.

5. Figure 2 seems too fragmented and feels like a series of disconnected details. Many of the genes in the small figures are actually the same, just with different functions. It is recommended to consolidate this information into a single, comprehensive figure that summarizes all the genes and their functions.

6. Figure 2 and the narrative in the article are inconsistent. The article is divided into two main sections: Function of IDD genes in SG5 lineage and Function of IDD genes in other lineages than SG5, each covering topics like leaf differentiation and disease resistance. However, Figure 2 presents genes related to leaf differentiation and other functions for both lineages together. It is recommended to consolidate Figure 2 into a single comprehensive figure for better clarity and persuasiveness.

Comments on the Quality of English Language

The writing throughout the document is strong and well-structured. However, a few details could be refined for clarity and consistency.

In lines 14-15, the sentence “Previous studies on Arabidopsis IDDs have demonstrated that AtIDD genes” can be revised to “Previous studies on Arabidopsis thaliana IDDs (AtIDDs) genes have demonstrated that these genes”

In line 19, the sentence IDDs in crops are involved in the regulation of can be revised to IDDs in crops are involved in regulating

Line 128: The font of "(k)" is different from the others.

Author Response

2024/09/21

Response to Reviewer 1

I am most grateful to the reviewers for the helpful comments on the original version of my manuscript. I have addressed all the comments made by the reviewers. I hope that the explanations and revisions of the work are satisfactory.

  • The last sentence of the abstract mentions that the study “offers fundamental insights for agricultural applications. (Line 22)” However, this aspect is barely covered in the article.

1: As the reviewer pointed out, this review itself does not offer the insights for agricultural applications. However, the findings which the review cover provide the ideas for agricultural applications. Therefore, the sentence was modified.

  • The article discusses the functions of IDD genes in different species. It is recommended to summarize these genes and their functions in a table, which would provide readers with a clearer understanding and also align with Figure 2.

2: As the reviewer suggested, I added the table which summarize the function of IDDs in crops.

  • Can you provide a phylogenetic tree of IDD protein from different species mentioned in the article? This would help readers better visualize the evolutionary relationships between these genes.

3: As the reviewer suggested, I also thought it is better to provide a phylogenetic tree. However, several previous research provide phylogenetic trees of IDDs which varies depending on researches (e.g. Prochetto, S. et al. annals of Botany 2020,126,85-101;Kumar, M. et al. Int J Mol Sci 2019, 20; Hirano, K. et al. The Plant Cell 2007, 19, 3058-3079; Yasumura,Y. et al. Current Biology 2007, 17, 1225-1230; Ishikawa, M. et al. Proceedings of the National Academy of Sciences 2023, 120,; Phokas, A and Coates, J.C., Evolution & Development 2021, 23, 137-154 ).

 Among them, the phylogenetic tree by Prochetto et al. (reverence [10]) which I referred to was created through detailed analysis and it is reliable. Unfortunately, I cannot make the same phylogenetic tree. To avoid the inaccuracies in the classification of IDD lineages, I do not provide a phylogenetic tree of IDDs. Instead, the lineages of IDDs proposed by Prochetto et al. were presented in the table1.

  • In the section “Function of IDD genes in SG5 lineage”, “leaf differentiation” is discussed first. However, in the section “Function of IDD genes in other lineages than SG5”, “leaf differentiation” is not addressed first. It is recommended to maintain consistency between the two sections. In the “Function of IDD genes in SG5 lineage” section, “Plant architecture” mainly refers to the tiller and leaf angle. In contrast, in the “Function of IDD genes in other lineages than SG5,” stem elongation is also considered part of plant architecture, but it is referred to specifically as “stem development.” For better logical consistency, it is advisable to use the same terminology and structure in both sections.

4: As the reviewer suggested, it may be better to align the order of the section in “Function of IDD genes in SG5 lineage” and “Function of IDD genes in other lineages than SG5”. However, the leaf differentiation in the part of “Function of IDD genes in other lineages than SG5” is about the function of maize NKD1 and this function is not the most well-known function (the most well-known function of NKD is the regulation of seed development). Therefore, I mentioned the function of NKD in seed development first and then leaf development. Although the section of “Stem elongation and secondary cell wall “ in the “Function of IDD genes in other lineages than SG5,”is not a part of “plant architecture”, it was placed first in the “Function of IDD genes in other lineages than SG5,” because it is related to plant architecture.

However, as the reviewer pointed out, the sections of “ leaf differentiation” and “ plant architecture” are intermixed. Therefore, I revise the section “2.1 leaf differentiation” because I mentioned about function of ZmIDD14 and 15 in plant architecture (because ZmIDD14 and 15 are ortholog of OsIDD12 and13, I put both of them in the same section). The function of ZmIDD14 and 15 in plant architecture was moved to the section “2.2 plant architecture”.

  • Figure 2 seems too fragmented and feels like a series of disconnected details. Many of the genes in the small figures are actually the same, just with different functions. It is recommended to consolidate this information into a single, comprehensive figure that summarizes all the genes and their functions.

5: As the reviewer pointed out, there are many figures and the same genes appeared in several figures. The figures are created to clearly illustrate the function of genes (regulation of the target genes and interaction with other proteins) because it is sometimes complicated. A single gene has various function and many genes function together with other gene. IDDs are transcription factors interacting with many other proteins and have several target genes. It would become very complicated figure If all of this information were to be summarized in a single figure. Therefore, I deliberately made separate figures.

  • Figure 2 and the narrative in the article are inconsistent. The article is divided into two main sections: “Function of IDD genes in SG5 lineage” and “Function of IDD genes in other lineages than SG5,” each covering topics like leaf differentiation and disease resistance. However, Figure 2 presents genes related to leaf differentiation and other functions for both lineages together. It is recommended to consolidate Figure 2 into a single comprehensive figure for better clarity and persuasiveness.

6: I understand the reviewer’s perspective. It is standard way to create figures that align with the context. However, I wanted to provide a comprehensive diagram to understand the function of IDD genes by grouping the figures based on gene functions. Moreover, although some IDD genes in SG5 lineage and in other lineages function together collaboratively, these functions are mentioned separately in the main text. Therefore, I tried to integrate the functions of these genes in a single figure (e.g. Figl 2 (e) ).

  • In lines 14-15, the sentence “Previous studies on Arabidopsis IDDshave demonstrated that AtIDD genes” can be revised to “Previous studies on Arabidopsis thaliana IDDs (AtIDDs) genes have demonstrated that these genes”

7: Thank you for the suggestion. The sentence was revised as suggested.

  • In line 19, the sentence “IDDs in crops are involved in the regulation of” can be revised to “IDDs in crops are involved in regulating”

8:  Thank you for the suggestion. The sentence was revised as suggested.

  • Line 128: The font of "(k)" is different from the others.

9: Thank you for pointing this out. The font was revised.

Reviewer 2 Report

Comments and Suggestions for Authors

Review of ijms-3180780

INDETERMINATE DOMAIN transcription factors in crops: plant architecture, disease resistance, stress response, flowering and more

Akiko Kozaki

This review describes the roles of IDDs in plant development, environmental responses, and evolution and presents potential agricultural applications. The review is thorough and comprehensive.  The English is good, but there are many minor mistakes.  I have flagged some but not all of them below.  It should therefore be carefully proofread before final submission

Lines 28-30: please rewrite for clarity and to correct grammar. It might be better to break them into 2 sentences.

Line 36:  please change “The IDD gene family encodes a transcription factor…” to “The IDD gene family encodes transcription factors…”

Line 39:  please change “…to which many of IDD proteins can bind…” to “…which many IDD proteins can bind…”

Line 42: please clarify what you mean by “after landing.”

Line 54:   please change “…which is the negative regulator...” to “…which are negative regulators...”

Line 66: please change “Function of IDD genes in SG5 lineage” to “Functions of IDD SG5 lineage genes”

Lines 92-98: please correct mistakes in spelling and capitalization

Line 104:  please change “counterparts” to “orthologs”

Line 125: please change “aspect” to “aspects”

Line 126: please change “Schematic diagram of IDD function” to “Schematic diagrams of IDD functions”

Line 127: please change “Fruits shape” to “Fruit shapes”

Line 130: please change “left-side line of the box indicate promoter” to “left-side lines of the boxes indicate the promoters”

Line 146: please change “affect” to “affects”

Lines 198- 202: please rewrite for clarity and to correct grammar.  I recommend separating into 2 sentences.

Lines 213-216: please rewrite for clarity and to correct grammar

Line 217:  please change to “Functions of IDD genes in lineages other than SG5”

Lines 239-241: please rewrite for clarity and to correct grammar

Lines 278-280 please rewrite for clarity and to correct grammar

Line 339: please change “regulate” to “regulates”

Line 342: please change “possess” to “possesses”

Line 364: please change “sequence” to “sites”

Line 368: please change “bind” to “bound”

Line 372: please change “regulate” to “regulated”

Line 389: please change to “…of the venation patterns in the inner leaf tissues…”

Line 395: please change to “…IDDs have been extensively analyzed in root development, GA synthesis, and signaling…”

Comments on the Quality of English Language

The English is good, but there are many minor mistakes.  I have flagged some but not all of them.  It should therefore be carefully proofread before final submission

Author Response

2024/09/21

Response to Reviewer 2

I am most grateful to the reviewers for the helpful comments and correction on the original version of my manuscript. I have addressed all the comments made by the reviewers. I hope that the explanations and revisions of the work are satisfactory.

  • Lines 28-30: please rewrite for clarity and to correct grammar. It might be better to break them into 2 sentences.

1: Thank you for pointing it out. The sentence was revised.

  • Line 36: please change “The IDD gene family encodes a transcription factor…” to “The IDD gene family encodes transcription factors…”

2: Thank you for pointing it out. The sentence was revised.

  • Line 39: please change “…to which many of IDD proteins can bind…” to “…which many IDD proteins can bind…”

3: Thank you for pointing it out. The sentence was revised.

  • Line 42: please clarify what you mean by “after landing.”

4: Thank you for pointing it out. The sentence was revised.

  • Line 54: please change “…which is the negative regulator...” to “…which are negative regulators...”

 5:  Thank you for pointing it out. The sentence was revised.

  • Line 66: please change “Function of IDD genes in SG5 lineage” to “Functions of IDD SG5 lineage genes”

 6: Thank you for pointing it out. The sentence was revised.

  • Lines 92-98: please correct mistakes in spelling and capitalization

 7: Thank you for pointing it out. The sentence was revised.

  • Line 104: please change “counterparts” to “orthologs”

8: Thank you for pointing it out. The word was revised.

  • Line 125: please change “aspect” to “aspects”

9: Thank you for pointing it out. The word was revised.

  • Line 126: please change “Schematic diagram of IDD function” to “Schematic diagrams of IDD functions”

10: Thank you for pointing it out. The word was revised.

  • Line 127: please change “Fruits shape” to “Fruit shapes”

 11: Thank you for pointing it out. The word was revised.

  • Line 130: please change “left-side line of the box indicate promoter” to “left-side lines of the boxes indicate the promoters”

12: Thank you for pointing it out. The word was revised.

  • Line 146: please change “affect” to “affects”

13: Thank you for pointing it out. The word was revised.

  • Lines 198- 202: please rewrite for clarity and to correct grammar. I recommend separating into 2 sentences.

14: Thank you for pointing it out. The sentence was revised.

  • Lines 213-216: please rewrite for clarity and to correct grammar

15: Thank you for pointing it out. The sentence was revised.

  • Line 217: please change to “Functions of IDD genes in lineages other than SG5”

 16: Thank you for pointing it out. The sentence was revised.

  • Lines 239-241: please rewrite for clarity and to correct grammar

 17: Thank you for pointing it out. The sentence was revised.

  • Lines 278-280 please rewrite for clarity and to correct grammar

 18: Thank you for pointing it out. The word was revised.

  • Line 339: please change “regulate” to “regulates”

 19: Thank you for pointing it out. The word was revised.

  • Line 342: please change “possess” to “possesses”

 20: Thank you for pointing it out. The word was revised.

  • Line 364: please change “sequence” to “sites”

 21: Thank you for pointing it out. The word was revised.

  • Line 368: please change “bind” to “bound”

 22: Thank you for pointing it out. The word was revised.

  • Line 372: please change “regulate” to “regulated”

 23: Thank you for pointing it out. The word was revised.

  • Line 389: please change to “…of the venation patterns in the inner leaf tissues…”

24:  Thank you for pointing it out. The word was revised.

  • Line 395: please change to “…IDDs have been extensively analyzed in root development, GA synthesis, and signaling…”

25: Thank you for pointing it out. The sentence was revised.